# Sol-Gel Synthesis of ZnO Nanoparticles Using Different Chitosan Sources: Effects on Antibacterial Activity and Photocatalytic Degradation of AZO Dye

**Ilham Ben Amor** [1,2,3]📷 , **Hadia Hemmami** [1,2]📷 , **Salah Eddine Laouini** [1,3] , **Mohammed Sadok Mahboub** [4] and **Ahmed Barhoum** [5,6,*]📷

1. Department of Process Engineering and Petrochemical, Faculty of Technology, University of El Oued, El Oued 39000, Algeria
2. Renewable Energy Development Unit in Arid Zones (UDERZA), University of El Oued, El Oued 39000, Algeria
3. Laboratory of Biotechnology Biomaterials and Condensed Materials, Faculty of Technology, University of El Oued, El Oued 39000, Algeria
4. LEVRES Laboratory, University of El Oued, El Oued 39000, Algeria
5. NanoStruc Research Group, Chemistry Department, Faculty of Science, Helwan University, Cairo 11795, Egypt
6. School of Chemical Sciences, Fraunhofer Project Centre, Dublin City University, D09 V209 Dublin, Ireland
* Correspondence: ahmed.barhoum@science.helwan.edu.eg

**Abstract:** Chitosan was used in the sol-gel synthesis of zinc oxide nanoparticles (ZnO NPs) as a capping agent in order to control the size, morphology, optical bandgap, photocatalytic efficiency, and antimicrobial activity. Different chitosan sources were used for the sol-gel synthesis of ZnO NPs, namely chitosan of shrimp shells, crab shells, and Streptomyces griseus bacteria. The photocatalytic efficiency was studied by using the methylene blue (MB) photodegradation test, and the antibacterial activity of the different types of ZnO NPs was investigated by the agar well diffusion technique. The particle size of ZnO NPs varied between 20 and 80 nm, and the band gap energy ranged between 2.7 and 3.2 eV. Due to the different chitosan sources, the ZnO NPs showed different antibacterial activity against Listeria innocua, Bacillus Subtiliis, Staphylococcus Aureus, Salmonella Typhimurium and Pseudomonas Aeruginosa. The ZnO NPs with lower band gap values showed better antibacterial results compared to ZnO NPs with higher band gap values. The MB dye removal of ZnO (shrimp shells), ZnO (crab shells), and ZnO (Streptomyces griseus) reached 60%, 56%, and 44%, respectively, at a contact time of 60 min, a low initial MB dye concentration of $6 \times 10^{-5}$ M, a solution temperature of 25 °C, and a pH = 7.

**Keywords:** green synthesis; metal oxide nanoparticles; photocatalytic degradation; AZO dye; photocatalytic activity; bandgap energy; antibacterial properties

## 1. Introduction

Zinc oxide nanoparticles have been synthesized and developed by materials scientists as a form of nanotechnology. Among the characteristics of ZnO NPs are their unique optical, electrical, photocatalytic, and antimicrobial properties, which enable the use of ZnO in various environmental and health applications [1]. ZnO NPs have low toxicity and are biodegradable, which means they are excellent nanocarriers for the delivery of various drugs [2–4]. ZnO NPs have a binding energy of 60 meV and an energy band of 3.37 eV, which gives them excellent chemical, electrical, and thermal stability. Their unique properties can be easily modified by changing the morphology of ZnO NPs through different synthetic routes, different starting materials, or modifiers used to fabricate the nanomaterial [5]. Due to its abundance, feasibility, and accessibility, it plays an essential role in many organic transformations. There are a variety of bond formation reactions in which zinc and

zinc oxide are used as excellent catalysts, such as C-N, C-O, and C-S [6]. Recently, ZnO NPs are gaining the attention of researchers due to their potential contribution to addressing environmental problems. However, efforts must be made to synthesize sunlight-responsive ZnO, which is only possible if the bandgap energy of the photocatalyst is reduced by doping with metals or nonmetals [7,8]. ZnO NPs have shown significant antibacterial activity against common food pathogens such as Listeria monocytogenes, *E. coli* O157:H7, *Campylobacter jejuni*, and *Salmonella spp.* proving their value as food preservatives [9]. ZnO NPs have been approved as safe by the Food and Drug Administration (FDA) [10].

Sol-gel synthesis of ZnO NPs has environmentally friendly aspects and various biomedical uses, and these techniques have rapidly evolved. Polysaccharides and biopolymer or plant extracts serve as modifying (capping) agents for the synthesis of biogenic ZnO NPs [11,12]. Among different modifiers, chitosan has gained attention for the synthesis of metal and metal oxide nanoparticles, including ZnO NPs [13,14]. Chitosan is an aminated polysaccharide commonly found in nature in crustaceans and insects. The extraction of chitosan can be carried out by four different methods and under different conditions. The main method for obtaining chitosan is based on the alkaline deacetylation of chitin with a strongly alkaline solution. Chitosan possesses functional groups, hydroxyl and amino groups, which are crucial for the chemical adsorption of metal ions [15]. Therefore, it is widely reported as a green capping agent (size control agent) in the synthesis of metal and metal oxide nanoparticles including ZnO, MgO, $TiO_2$, Au, Ag, and Cu [16]. Previous studies have shown that functionalization of the surface of metallic nanoparticles by chitosan offers many advantages, including the improvement of optical properties, drug loading and release, and enhancement of antimicrobial activity.

Considering the superior properties of chitosan, chitosan was used as a capping agent in the sol-gel synthesis of ZnO NPs to control the size, morphology, optical band gap, photocatalytic efficiency, and antimicrobial activity. The chitosan source and extraction process determine the molecular weight and degree of deacetylation of the chitosan, which affects the properties of the obtained nanoparticles. In this study, ZnO NPs were successfully prepared from aqueous zinc chloride ($ZnCl_2$) and various types of chitosan (sources: shrimp shells, crab shells, and streptomyces griseus). We used $ZnCl_2$, NaOH and chitosan as non-toxic reagents and water as solvent. The prepared ZnO NPs were characterized by X-ray powder diffraction (XRD), scanning electron microscopy (SEM) and ultraviolet spectroscopy (UV-Vis). The effects of different chitosan sources on the photocatalytic activity of ZnO for the removal of methylene blue (MB) from aqueous solution and as a bactericidal agent against Gram-negative and Gram-positive bacteria were investigated. The synergistic effect of chitosan leads to an increase in antibacterial activity and photocatalytic efficiency.

## 2. Results and Discussion

Chitosan was considered an effective biopolymer for the preparation of ZnO NPs in this study because of its ability to bond with zinc ions through the amino and hydroxyl groups [17]. The $Zn^{2+}$ ions are hydrolyzed forming $Zn(OH)_2$ ($Zn^{2+} + 2\,OH \rightarrow Zn(OH)_2$) and bind to chitosan when chitosan and $ZnCl_2$ are mixed. Stable chitosan/$Zn(OH)_2$ nanoclusters are formed when the pH of the solution is raised to pH = 9 by adding NaOH dropwise [18]. The chitosan molecules act as a steric barrier with a positive charge density covered around the $Zn(OH)_2$ nanoclusters. The steric barrier between $Zn(OH)_2$ NPs allowed the formation of homogeneous nanoparticle solutions. Drying at 100 °C and subsequent annealing at 500 °C in air results in ZnO NPs with high crystallinity and improved optical properties. Thermal conversion of $Zn(OH)_2$/chitosan to ZnO NPs is a complicated process involving two or possibly three stages. The first stage occurs between 30 and 150 °C and is due to the thermal elaboration of water and the formation of ZnO xerogel. The second stage between 150 and 340 °C is the thermal decomposition (oxidation) of chitosan. At (500 °C), the chitosan was completely removed and crystallized ZnO NP was formed. Figure 1 shows the sol-gel synthesis of ZnO NPs using different types of

chitosan and their application in the antibacterial and photocatalytic degradation of MB dye.

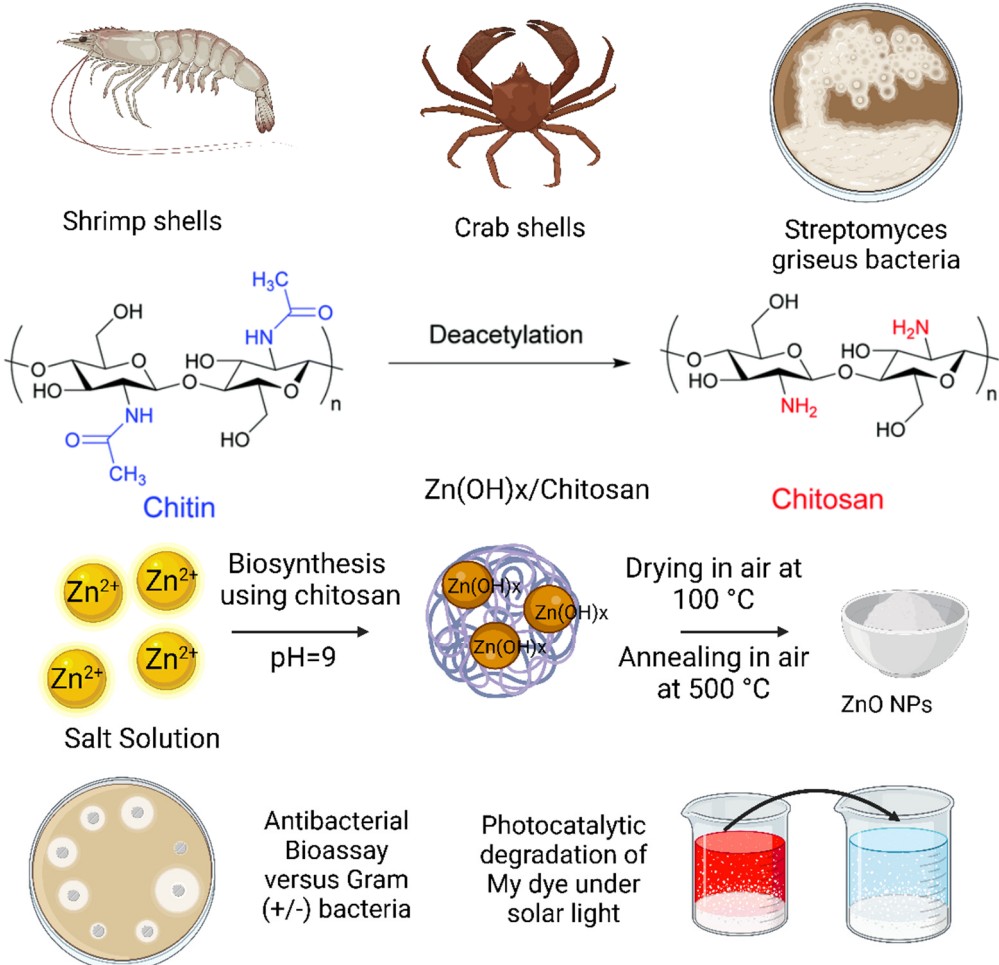

**Figure 1.** The schematic representation of the sol-gel synthesis of ZnO NPs using different types of chitosan (sources: *shrimp shells*, *crab shells*, and *Streptomyces griseus bacteria*) and their application in antibacterial and photocatalytic degradation of MB dye.

### 2.1. Crystallinity and Crystallite Size

Figure 2 shows the XRD pattern of ZnO NPs synthesized from chitosan of *shrimp shells*, *crab shells*, and *Streptomyces griseus bacteria*. Compared to the JCPDS cards, the diffraction peaks labeled (100), (002), (101), (102), (110), (103), (200), (112), and (201) are in a good fit to the hexagonal ZnO wurtzite structure (JCPDS card no: 01−079−0205) [19]. No other peaks are observed, indicating the high purity and crystallinity of synthesized ZnO NPs. The Debye-Scherer equation, $D = K\lambda/(\beta\cos\theta)$, was used to calculate the size of ZnO NPs' crystallites, where D is the average crystallite size, K is the form factor (0.9), $\lambda$ is of the wavelength of X-ray (0.15418 nm, CuK$\alpha$), $\beta$ is the maximum width at full width (FWHM) and $\theta$ is the Bragg angle [20]. Table 1 shows the XRD calculations (crystallite size and lattice parameters) of prepared ZnO NPs. The crystallite size of ZnO NPs varied between 30.9 to 35.8 nm. Different properties of chitosan (capping agent) lead to a different degree of stabilization of $Zn(OH)_2$ NPs against agglomeration. This leads to the formation of ZnO NPs with the observed variation in crystallite size.

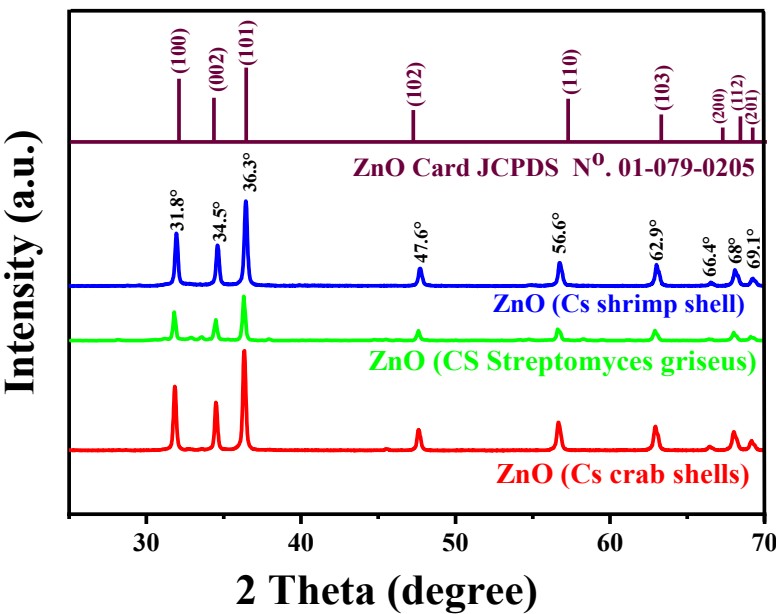

**Figure 2.** The XRD diffraction pattern of the synthesized ZnO NPs from chitosan of shrimp shells, crab shells and Streptomyces griseus bacteria.

**Table 1.** The XRD calculations of ZnO NPs from chitosan of *shrimp shells*, *crab shells*, and *Streptomyces griseus bacteria*.

| Chitosan Source | 2θ (°) | FHWM | Crystallite Size (nm) | Lattice Parameter (Å) | |
|---|---|---|---|---|---|
| | | | | A | C |
| ZnO NPs by CS of *shrimp shells* | 36.3 | 0.2657 | 30.9 | 3.25 | 5.19 |
| ZnO NPs by CS of *crab shells* | 36.5 | 0.253 | 33.6 | 3.23 | 5.17 |
| ZnO NPs by CS of *Streptomyces griseus bacteria* | 36.3 | 0.2423 | 35.8 | 3.24 | 5.19 |

### 2.2. Morphological Investigation

SEM was used to examine the morphology of synthesized ZnO NPs from different chitosan (see Figure 3). The ZnO NPs were characterized by random spherical shapes (Figure 3a,c,e). The reaction parameters including the concertation of $ZnCl_2$, source and amount of chitosan, temperature, and pH, affect the shape and particle size of the ZnO NPs [21]. Najoom et al. [22] observed similar morphologies. The particle size distribution histograms show indicated that the average size distribution of the ZnO NPs by *crab shells* was around 30–80 nm (Figure 3b), 20–70 for ZnO by *shrimp shells* (Figure 3d), and 35–75 nm for ZnO NPs by *Streptomyces griseus bacteria* (Figure 3f). Different sources of chitosan lead to a different degree of stabilization of $Zn(OH)_2$ NPs against agglomeration. This leads to the formation of ZnO NPs with the observed variation in particle sizes. The crystallite size is smaller than the particle size determined by the SEM analysis. This may indicate that the prepared ZnO NPs are polycrystalline. According to the SEM/EDS elemental analysis (Table 2), O and Zn were present in the ZnO NPs of *shrimp shells* with an atomic fraction of 52.01% and 47.99%, respectively, and in the ZnO NPs of *crab shells* with an atomic fraction of 52.59% and 34.16%, respectively. According to ZnO NPs of *Streptomyces griseus bacteria*, the atomic masses of O and Zn in the ZnO NPs were 26.80% and 51.13%, respectively.

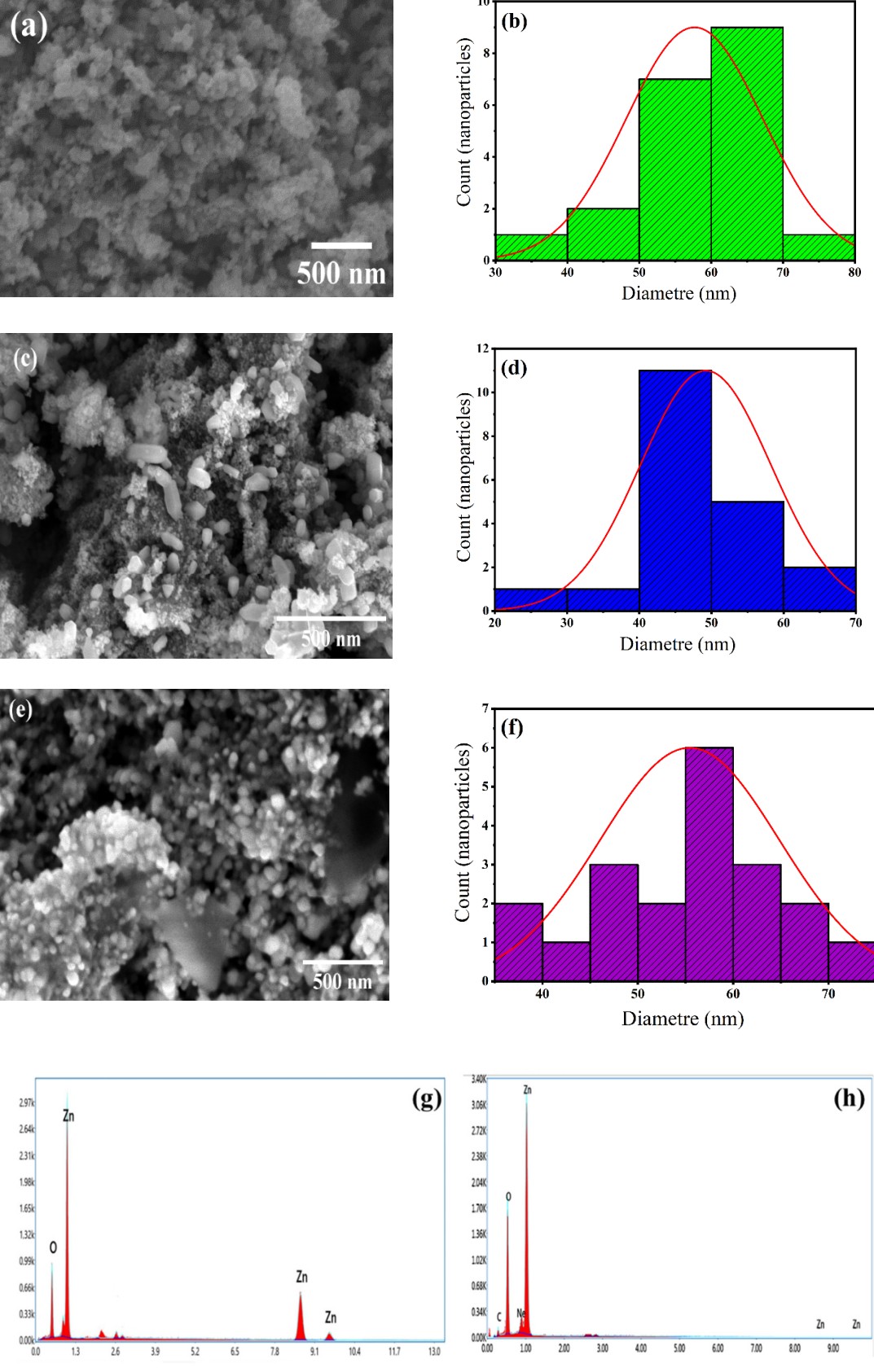

**Figure 3.** *Cont*.

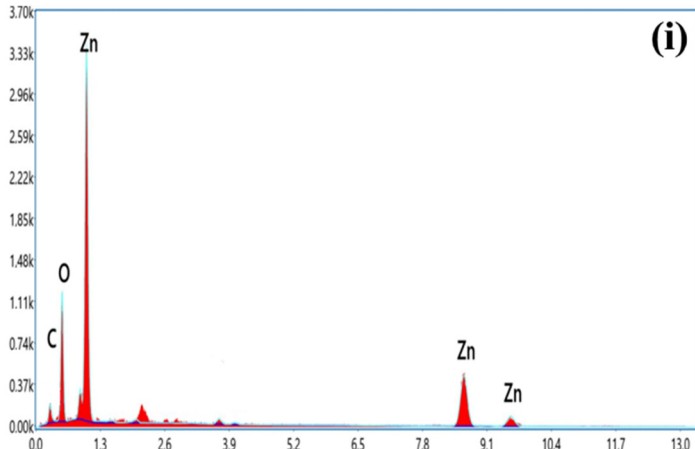

**Figure 3.** SEM-EDX image and particle size distributions of ZnO NPs from chitosan of (**a**,**b**,**g**) *crab shells*, (**c**,**d**,**h**) *shrimp shells* and (**e**,**f**,**i**) *Streptomyces griseus bacteria*.

**Table 2.** The elemental composition of the ZnO NPs.

| Compound | Composition of ZnO | |
|---|---|---|
| | Element | Atomic percentage % |
| ZnO NPs by *shrimp shells* | O K | 52.01 |
| | Zn K | 47.99 |
| | Totals | 100 |
| ZnO NPs by *crab shells* | C K | 12.43 |
| | O K | 52.59 |
| | Ne K | 0.82 |
| | Zn K | 34.16 |
| | Totals | 100 |
| ZnO NPs by *Streptomyces griseus bacteria* | C K | 22.07 |
| | O K | 51.13 |
| | Zn K | 26.8 |
| | Totals | 100 |

### 2.3. UV-Vis Spectroscopy Analysis

ZnO NPs are well-known photocatalysts that are generally active in UV light due to their wide band gap. The efficiency of ZnO NPs strongly depends on their electronic band structure, UV-Vis absorption, and band gap energy. It has been reported that the generated optical band gaps of ZnO nanostructures range from 2.7 eV to 4.7 eV for ZnO NPs formed in different solvent systems [23,24]. The bandgap energy for an efficient photocatalyst should be less than 3 eV to extend light absorption into the visible range and efficiently utilize solar energy. The UV-Vis spectra of the ZnO NPs synthesized from different chitosan sources are shown in Figure 4a. The UV-Vis spectra of the ZnO NPs synthesized from different chitosan sources (crab shells, crab shells and Streptomyces griseus bacteria) show distinct absorption bands at 343, 330 and 328 nm, respectively. The absorption spectrum of the ZnO NPs is in agreement with that of Vijayakumar et al. [25] and Vaseem et al. [26]. The band gap energy of synthesized ZnO NPs was calculated by plotting $(hv)^2$ versus energy (eV) as shown in Figure 4b. The band gap energies of ZnO NPs synthesized from chitosan of shrimp shells, crab shells and Streptomyces griseus bacteria are 2.73, 2.87 and 3.21 eV, respectively. These results confirm that the use of chitosan during the ZnO NPs synthesis

has significantly affected the optical properties of the ZnO NPs and reduced the bandgap energy to a minimum.

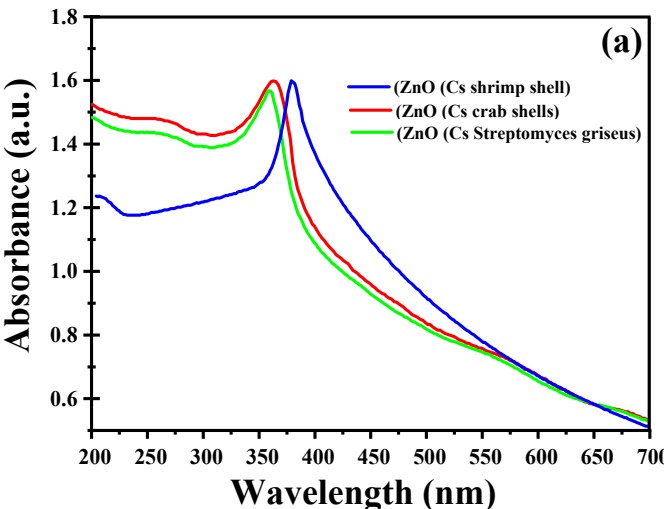

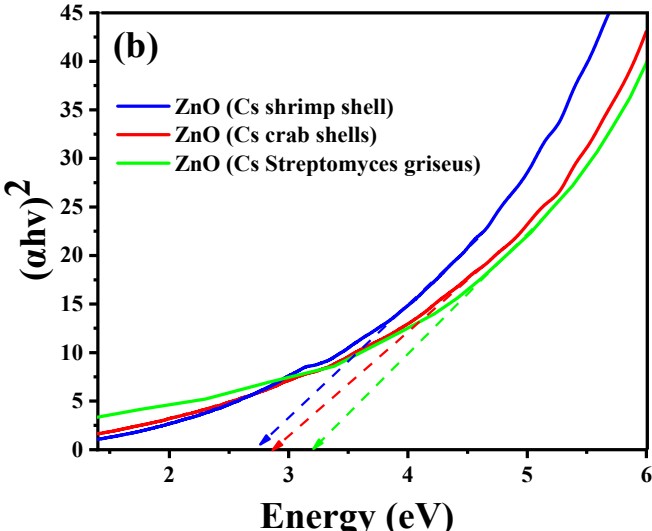

**Figure 4.** The optical properties of ZnO NPs from *chitosan of shrimp shells, crab shells,* and *Streptomyces griseus bacteria*. (**a**) UV-Visible spectrum, and (**b**) optical energy bandgap.

### 2.4. Antibacterial Activities

ZnO NPs exhibit attractive antibacterial properties because the specific surface area increases as the particle size decrease, resulting in increased reactivity of the particle surface. Zinc ions ($Zn^{2+}$) released from ZnO NPs exhibit antimicrobial activity against various strains of bacteria and fungi. ZnO NPs combine two additional mechanisms of antimicrobial activity, namely reactive oxygen species generation and direct contact with cell walls [27]. The generation of reactive oxygen species leads to the leakage of reducing sugars, DNA, and proteins from the membrane and reduces cell viability. Figure 5 and Table 3 show the antibacterial activity of ZnO NPs against Gram-positive bacteria such as *Bacillus subtilis*, *Staphylococcus aureus*, and *Listeria innocua*, and against Gram-negative bacteria such as *Pseudomonas aeruginosa* and *Salmonella typhimurium*. Figure 5 shows that the antibacterial effect on Gram-positive bacteria is much stronger for the different types of ZnO NPs. Gram-positive bacteria have a simple cytoplasmic membrane with a multilayer peptidoglycan polymer layer and a weaker cell wall, while Gram-negative bacteria have a thicker cell wall and are consequently more resistant to ZnO NPs [28]. In contrast, the wall of a Gram-

negative bacterium consists of two cell membranes, those being an outer membrane and a plasma membrane covered with a thin peptidoglycan layer [27].

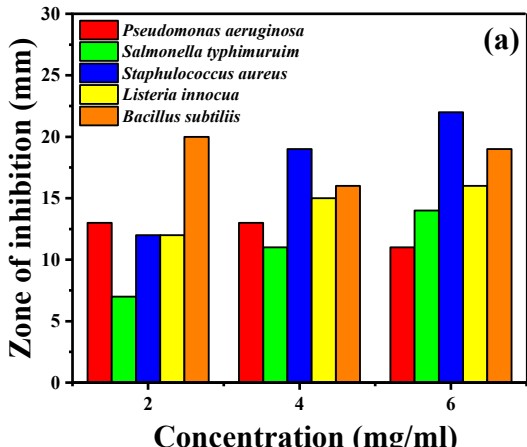

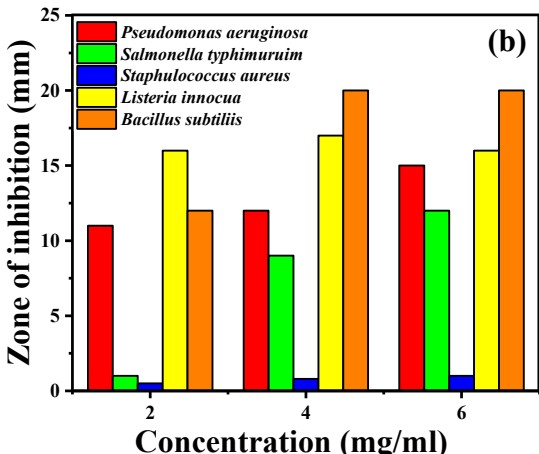

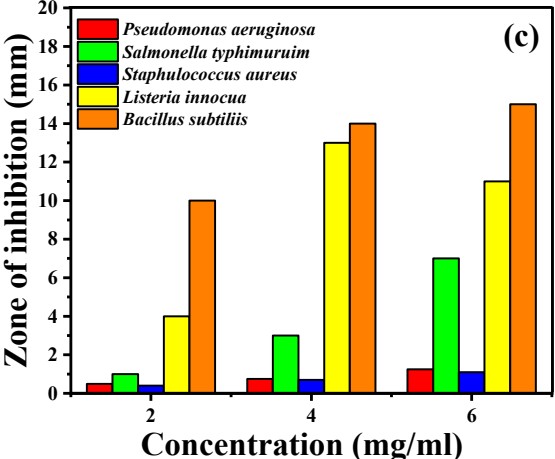

**Figure 5.** The antibacterial activity of ZnO NPs from chitosan of (**a**) *shrimp shells*, (**b**) *crab shells* and (**c**) *Streptomyces griseus bacteria* at various concentrations against different bacteria.

**Table 3.** The zone of inhibition of ZnO NPs from chitosan of *shrimp shells*, *crab shells*, and *Streptomyces griseus bacteria*.

| Sample | Conc. | Zone of Inhibition [a] (mm) | | | | |
|---|---|---|---|---|---|---|
| | | Gram-Negative | | Gram-Positive | | |
| | | *Pseudomonas aeruginosa* | *Salmonella typhimuruim* | *Staphulococcus aureus* | *Listeria innocua* | *Bacillus subtiliis* |
| ZnO NPs by CS of *shrimp shells* | 2 mg/mL | 13 ± 0.25 | 7 ± 0.15 | 12 ± 0.10 | 12 ± 0.12 | 20 ± 0.30 |
| | 4 mg/mL | 13 ± 0.20 | 11 ± 0.15 | 19 ± 0.30 | 15 ± 0.15 | 16 ± 0.35 |
| | 6 mg/mL | 11 ± 0.15 | 14 ± 0.2 | 22 ± 0.35 | 16 ± 0.19 | 19 ± 025 |
| ZnO NPs by CS of *crab shells* | 2 mg/mL | 11 ± 0.10 | 1 ± 0.05 | 0.5 ± 0.05 | 16 ± 0.2 | 12 ± 0.35 |
| | 4 mg/mL | 12 ± 0.15 | 9 ± 0.15 | 0.8 ± 0.05 | 17 ± 0.25 | 20 ± 0.1 |
| | 6 mg/mL | 15 ± 0.25 | 12 ± 0.25 | 1 ± 0.1 | 16 ± 0.3 | 20 ± 0.2 |
| ZnO NPs by CS of *Streptomyces griseus bacteria* | 2 mg/mL | 0.5 ± 0.15 | 1 ± 0.1 | 0.4 ± 0.12 | 4 ± 0.15 | 10 ± 0.20 |
| | 4 mg/mL | 0.75 ± 0.05 | 3 ± 0.1 | 0.7 ± 0.04 | 13 ± 0.2 | 14 ± 0.25 |
| | 6 mg/mL | 1.25 ± 0.05 | 7 ± 0.15 | 1.1 ± 0.05 | 11 ± 0.20 | 15 ± 0.30 |
| ciprofloxacin (CIP-5) | 50 μg | 22 ± 0.4 | 17 ± 0.15 | 14 ± 0.2 | 24 ± 0.3 | 24 ± 0.2 |

[a] average value of three readings.

According to Padmavathy et al. [29], ZnO NPs also have an abrasive surface roughness that impairs the antibacterial process and destroys the bacterial membrane of both Gram-positive and Gram-negative bacteria. However, the particle size has a significant effect on the antibacterial activity of ZnO. It has been reported that the antibacterial activity of ZnO is increased by decreasing the particle size [30], which was confirmed in our study: the smaller the particle size, the higher the antibacterial activity. Compared with our study and according to the reports, the antibacterial activity of ZnO is increased when the particle size is decreased [31]. Jones et al. [30] investigated the antibacterial activity of ZnO, $TiO_2$, CuO, $CeO_2$, $Al_2O_3$, and MgO against the bacterium *S. aureus* and compared their antibacterial activities. The ZnO NPs among them showed significant growth inhibition [30]. Yamamoto et al. [32] investigated the ability of *E. coli* and *S. aureus* to control bacteria cultured in an infusion medium to the influence of ZnO NPs size. ZnO particles were prepared in the following particle sizes of 0.1, 0.2, 0.3, 0.5, and 0.8 μm by heating reagent grade ZnO powder to 1400 °C followed by planetary ball milling. It was found that the antibacterial activity increased with decreasing particle size.

*2.5. Photocatalytic Ability to MB Degradation*

Photocatalysis, a simple process, and an environmentally friendly technology can degrade organic pollutants contained in wastewater to water. ZnO NPs are photocatalysts that can degrade hazardous organic compounds when exposed to sunlight [33]. The valuable photocatalytic mechanism depends on the properties of ZnO NPs and the configuration of the active species generated in the reaction medium. Radical scavenging experiments were performed with ZnO NPs to investigate the probable contribution of different active species to the photocatalytic degradation of AZO dyes. It was found that radical scavengers such as isopropanol, formic acid, oxalic acid, and ascorbic acid play an important role in the degradation of AZO dye under solar irradiation [34]. However, no radical scavengers were used in this study. In this work, MB was employed as a model organic pollutant to assess the photocatalytic activity of ZnO NPs under UV light irradiation. MB is a toxic, carcinogenic, and non-biodegradable type of AZO dye that causes serious risks to human health and the environment [35]. Therefore, there is a need to develop an environmentally friendly, efficient technology to remove MB from wastewater. Among the various processes, photocatalytic degradation is recommended to remove MB from wastewater. It has the advantage of completely mineralizing the dye into simple and non-toxic species

and reducing processing costs. The difference between the concentration of the dye MB in the aqueous solution before and after the photodegradation test was used to quantify the amount of adsorption [18]:

$$q_e = \frac{(C_0 - C_e)V}{m} \tag{1}$$

The equilibrium concentration of the dye on the adsorbent is represented by $q_e$ (mg·g$^{-1}$). The initial and equilibrium concentrations of the dye solution are $C_0$ and $C_e$ (mg·L$^{-1}$), respectively. In this case, m is the weight of ZnO used (g), and V is the volume of the dye solution (L). UV-vis spectroscopy is used to observe the stimulation (Figure 6). Once ZnO NPs are introduced as components in the reaction mixture, the dye is catalytically reduced. Equation (2) was used to calculate the photocatalytic degradation efficiency (%) of the dye MB by the ZnO adsorbent [36,37]:

$$Degradation\ ratio\ (\%) = \frac{(C_0 - C_t)}{C_0} \times 100 \tag{2}$$

where $C_t$ is the current concentration and $C_0$ is the starting concentration of MB. Figure 6 shows that the ZnO NPs synthesized with *shrimp shells* adsorb and photodegrade MB better over time than the ZnO NPs synthesized with *crab shells* and *Streptomyces griseus bacteria*, which explains that the chitosan source affects the formation of ZnO NPs. Using optimal experimental conditions, 60%, 56%, and 44% degradations were achieved for *shrimp shells*, *crab shells*, and *Streptomyces griseus bacteria*, respectively, within 60 min (see Figure 6). Table 4 shows a comparison of the AZO dye removal efficiency results of ZnO NPs prepared from different chitosan sources with this work. The ZnO NPs prepared in this work have high dye removal efficiency in the short time of 60 min compared to other studies.

According to Equations (3)–(9), the photoexcitation, charge separation and migration, and finally, oxidation-reduction reactions at the surface, the reactive species formed during the irradiation of ZnO NPs are (VB), h$^+$, $O_2^-$, and OH [38]. The ZnO NPs are exposed to solar radiation (Figure 7), which causes electrons to move from VB to CB. The resulting energy is larger than the band gap of ZnO (2.37, 2.87, 3.21 eV), which promotes the formation of conduction band electrons (e) and valence band holes (h$^+$). The photogenerated holes on the VB could either directly oxidize the adsorbed MB dye or directly react with hydroxyl (OH·). Meanwhile, oxygen ($O_2$) adsorbed on the surface of ZnO NPs can be converted into superoxide radicals ($O_2$) by photoelectrons at the CB. For this reason, both OH· and the generated $O_2$ can photocatalytically degrade the MB dye [39,40].

$$Zno \overset{h\gamma}{\rightarrow} ZnO(e^-(CB)) + ZnO(h^+(VB)) \tag{3}$$

$$ZnO(h^+(VB)) + H_2O \rightarrow ZnO + H^+ + OH^+ \tag{4}$$

$$ZnO(e^-(CB)) + O_2 \rightarrow ZnO + O_2^{-\bullet} \tag{5}$$

$$H^+ + O_2^{-\bullet} \rightarrow OH_2^{\bullet} \tag{6}$$

$$OH_2^{\bullet} + OH_2^{\bullet} \rightarrow H_2O_2 + O_2 \tag{7}$$

$$H_2O_2 \overset{h\gamma}{\rightarrow} 2OH^{\bullet} \tag{8}$$

$$Methylene\ blue + OH^{\bullet} \rightarrow Degradation\ products + CO_2 + H_2O \tag{9}$$

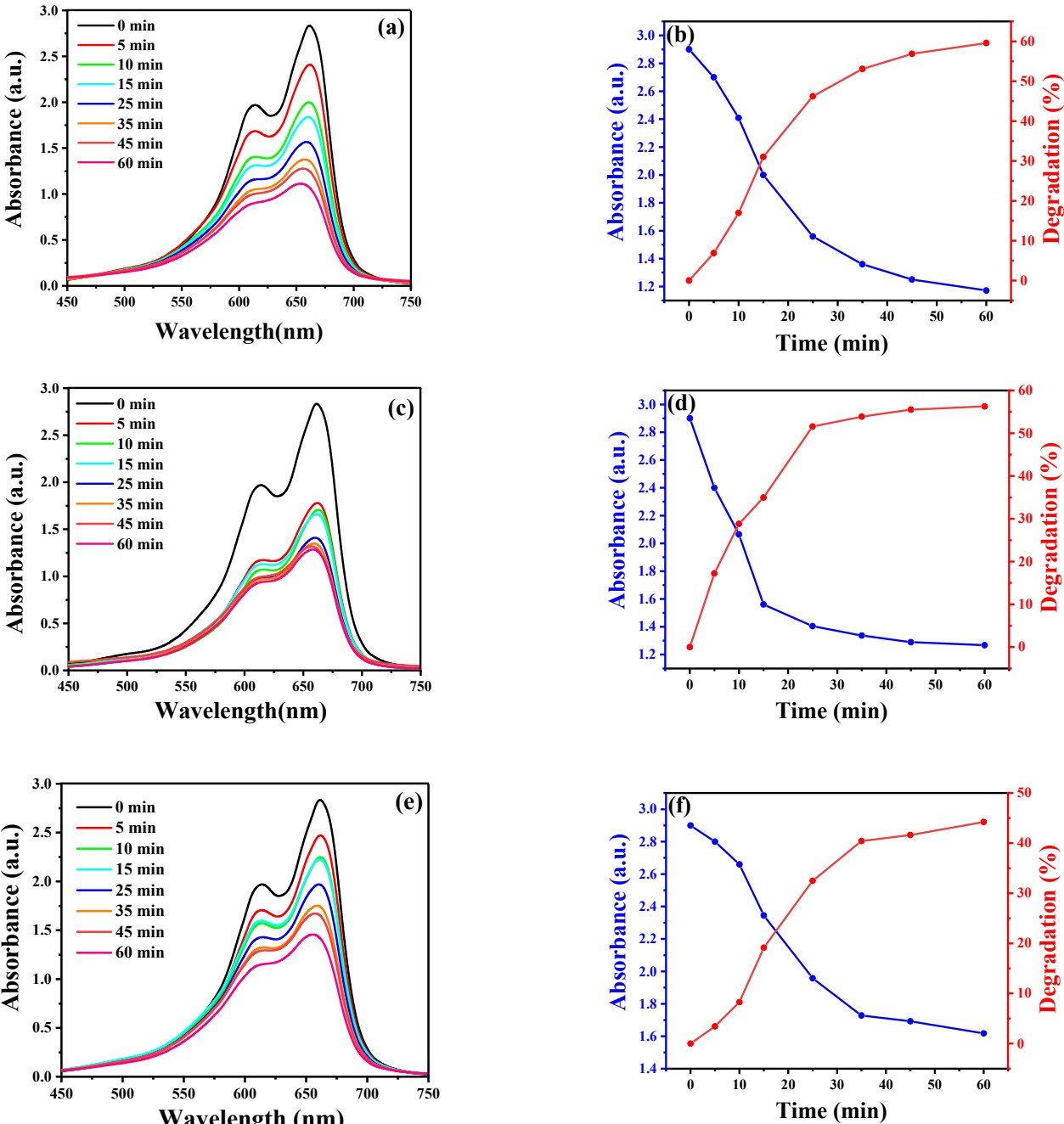

**Figure 6.** The time influence reaction of ZnONPs on the degradation of MB dye: (**a**) *shrimp shells*, (**c**) *crab shells*, (**e**) *Streptomyces griseus bacteria*. Degradation percent of MB dye (**b**) ZnO NPs by *shrimp shells*, (**d**) ZnO NPs by *crab shells*, and (**f**) ZnO NPs by *Streptomyces griseus bacteria*.

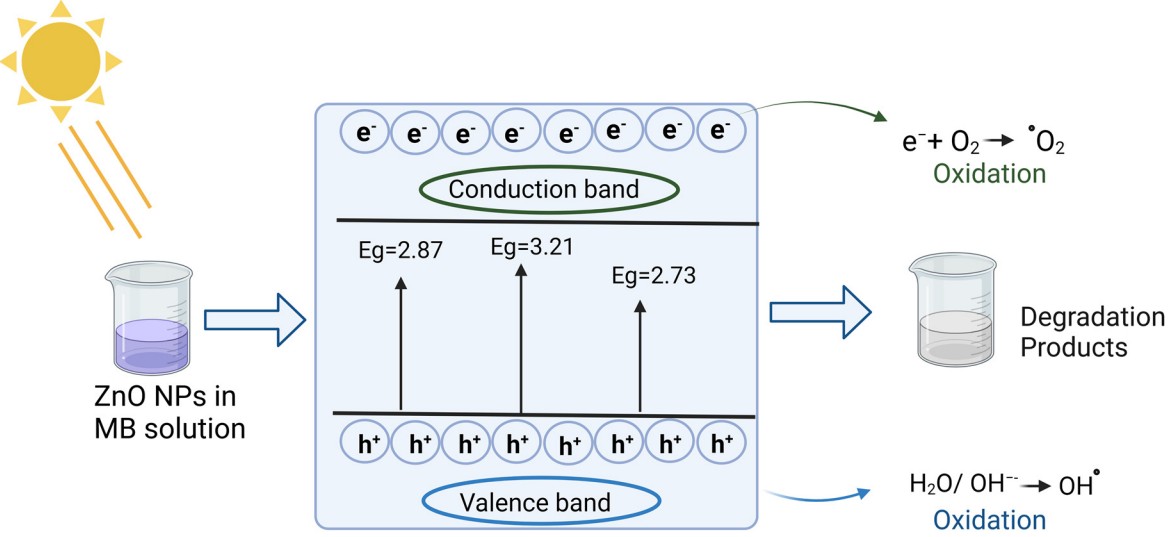

**Figure 7.** The schematic presentation showing the mechanism by which ZnO NPs are used to photodegrade the MB dye.

**Table 4.** The comparison of results of the AZO dye removal efficiency of ZnO NPs prepared from different growth controlling agents compared to this work.

| Modifier | AZO Dye | Time (min) | Dye Removal (%) | Ref. |
|---|---|---|---|---|
| *Extract of Becium grandiflorum* | Methylene Blue | 60 | 30 | [41] |
| *Ruellia tuberosa extract* | Malachite green (MG) | 60 | 59 | [42] |
| *Myrica esculenta fruits extract* | Methylene Blue | 60 | 29 | [43] |
| *leaf extract of the plant Ruta Chalepensis* | Methyl Red | 60 | 74 | [44] |
| *Ulva lactuca seaweed extract* | Methylene Blue | 60 | 45 | [45] |
| *Chitosan of shrimp shells* | Methylene Blue | 60 | 60 | |
| *Chitodan of crab shells* | Methylene Blue | 60 | 56 | This work |
| *Chitosan of Streptomyces griseus bacteria* | Methylene Blue | 60 | 44 | |

## 3. Experimental

### 3.1. Materials

Three types of chitosan were obtained from Aldrich (Aldrich Co., Steinheim, Germany), those being chitosan I (powder, shrimp shells, product number 50494), chitosan II (powder, crab shells, product number 48165), and chitosan III (powder, Streptomyces griseus, product number C9830). Dimethyl sulfoxide (DMSO, 99%), zinc chloride (ZnCl$_2$, 99%), hydrochloric acid (HCl, 99%), hydrogen peroxide (H$_2$O$_2$, 98%), sodium hydroxide (NaOH, 97%), methylene blue (C$_{16}$H$_{18}$ClN$_3$S, 82%), and acetic acid (CH$_3$COOH, 99.5%) were acquired from Biochem Chemophara. Additionally, Mueller-Hinton agar was made by the Algerian business Bioscan Industrie.

### 3.2. Sol-Gel Synthesis of ZnO NPs Using Different Chitosan Sources

In order to prepare ZnO NPs using three sources of chitosan, a 1.0% ($w/v$) chitosan solution was prepared by dissolving in a 1% ($v/v$) acetic acid solution. After complete dissolution, 50 mL of a solution containing 0.1 M ZnCl$_2$ and 5 mL of chitosan was prepared. The pH of the mixtures was adjusted by adding 0.1 M NaOH and stirring continuously until the pH became basic (pH 9–10) at 60 °C for 4 h. The precipitate was then centrifuged at 2500× $g$ rpm, rinsed numerous times with deionized water, and then dried at 100 °C for 3 h and annealed for 4 h at 500 °C, to acquire ZnO NPs.

### 3.3. Physicochemical Characterization of ZnO NPs

The absorption spectra of ZnO NPs were recorded through ultraviolet spectroscopy (UV-2450, Shimadzu, Duisburg, Germany). The samples were investigated by dispersing 0.1 mg of them in 2 mL of distilled water. The Tauc relationship $(hv) = A (h − Eg)^n$ [11] was used to determine the band gap energy (Eg). The crystalline structure of the ZnO NPs was investigated by X-ray diffraction (Rigaku D/Max-2000, Tokyo, Japan). A Fourier transform infrared spectrophotometer (Perkin-Elmer Corporation, Series 1725x, Norwalk, CT, USA) was used to identify the functional groups present and the size of ZnO NPs. A scanning electron microscope (FESEM, Leo Supra 55, Zeiss Inc., Oberkochen, Germany) was used to examine the morphology and elemental composition of the samples.

### 3.4. Bioassay for Antibacterial

The antibacterial activity of ZnO NPs against a range of bacterial species, including *Bacillus subtilis* (ATCC6633), *Pseudomonas aeruginosa* (ATCC9027), *Staphylococcus aureus* (ATCC6538), *Salmonella typhimurium* (ATCC14028), and *Listeria innocua* (CLIP74915), was investigated using the agar well diffusion method. About 100 L of a broth culture of bacterial strains matured for 24 h was prepared and distributed on culture plates. Agar wells with a diameter of 6 mm were made in each of the agar plates using a sterile stainless steel cork borer. The bactericidal activity of the ZnO NPs was tested using different doses of 2, 4, and 6 $mg.mL^{−1}$ in DMSO. Antibacterial assay of the samples was performed with the reference drug ciprofloxacin (CIP-5) after incubating the plates at 37 °C for 24 h.

### 3.5. Photocatalytic Degradation

The photocatalytic activity of ZnO NPs was assessed by observing the photochemical deterioration of MB dye in an aqueous solution under a 1000 W ultraviolet (UV) radiation lamp [39,46]. Before illumination, the MB dye ($6 \times 10^{−5}$ M) was mixed with the proper amount of catalyst (30 mg) for 15 min in the dark to absorb the largest amount of MB. A UV-Vis spectrometer is then used to track the development of the reaction at intervals of (0, 5, 10, 15, 25, 35, 45, and 60 min). Under UV illumination, the experiment of a complete reduction reaction was carried out. The intensity of the blue color of the reaction mixture decreased steadily. In order to stop the degradation, the solution was centrifuged. $\lambda_{max}$ = 663 nm, the absorbance of the solution was measured [17].

## 4. Conclusions

Chitosan is an abundant biopolymer with chemical functional groups that can be modified for a variety of applications, including as a surface modifier and growth control agent in the synthesis of nanoparticles. In this work, ZnO NPs were prepared by sol-gel synthesis method using different chitosan sources (shrimp shells, crab shells, and Streptomyces griseus bacteria) as a growth controlling agent (capping agent). The particle size of ZnO NPs from shrimp shells, crab shells and Streptomyces griseus bacteria are 30.9 nm, 33.5 nm and 35.9 nm, and the band gap energies are 2.7, 2.9 and 3.2 eV, respectively. Different chitosan sources lead to different degrees of stabilization of ZnO NPs against agglomeration. This leads to the formation of ZnO NPs with the observed variations in crystallite sizes, particle sizes, and band gap energies and enhancement of photocatalytic and antibacterial activity. The obtained ZnO NPs showed good antibacterial activity against Gram-positive bacteria and were significantly more resistant than Gram-negative bacteria. Direct contact with cell walls, release of $Zn^{2+}$ ions from the ZnO NPs and generation of reactive oxygen species causes good antimicrobial activity against Listeria innocua, Bacillus Subtiliis, Staphylococcus Aureus, Salmonella Typhimurium and Pseudomonas Aeruginosa. The prepared ZnO NPs achieved good dye degradation efficiencies of 60%, 56% and 44% for shrimp shells, crab shells and Streptomyces griseus bacteria, respectively, within 60 min at a solution temperature of 25 °C and pH of 7. The method proved to be new, simple, efficient and fast.

**Author Contributions:** Conceptualization, I.B.A., H.H., S.E.L., M.S.M. and A.B.; methodology, I.B.A., H.H., S.E.L., M.S.M. and A.B.; software, I.B.A., H.H., S.E.L., M.S.M. and A.B.; validation, I.B.A., H.H., S.E.L., M.S.M. and A.B.; formal analysis, I.B.A., H.H., S.E.L., M.S.M. and A.B.; investigation, I.B.A., H.H., S.E.L., M.S.M. and A.B.; data curation, I.B.A., H.H., S.E.L., M.S.M. and A.B.; writing—original draft preparation, I.B.A., H.H., S.E.L., M.S.M. and A.B.; writing—review and editing, I.B.A., H.H., S.E.L., M.S.M. and A.B.; visualization, I.B.A., H.H., S.E.L., M.S.M. and A.B.; supervision, S.E.L. and A.B.; project administration, S.E.L. and A.B.; funding acquisition, S.E.L. and A.B. All authors have read and agreed to the published version of the manuscript.

**Funding:** Ahmed Barhoum (NanoStruc Research Group at Helwan University, Projects PI) would like to thank Joint Egyptian Japanese Scientific Cooperation (JEJSC, Project No. 42811, 2021–2022).

**Data Availability Statement:** All data of this work described in this article.

**Conflicts of Interest:** The authors declare that they have no known competing financial interests or personal relationships that could have appeared to influence the work reported in this paper.

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
