# Peer review of "Sol-Gel Synthesis of ZnO Nanoparticles Using Different Chitosan Sources: Effects on Antibacterial Activity and Photocatalytic Degradation of AZO Dye"

_catalysts, doi:10.3390/catal12121611_

Round 1

Reviewer 1 Report

The subject manuscript "Biosynthesis of zinc oxide nanoparticles using different chitosan sources: Effects on photocatalytic degradation of AZO dye and antibacterial activity" was evaluated and considered for publication in the Catalysts subject to the following revision. 

1....The introduction is very poorly written, author need to discuss the importance of this study, authors can take help from the following studies, e.g., Chemical Engineering Journal 351 (2018) 841-855: Scientific Reports 10 (2020), 2042; J. Clean. Prod. 241 (2019) 118263.

2....Why did the author select green synthesis method for the synthesis of ZnO, though it is very tedious, time consuming, and results into low yield of ZnO.

3....What was the intensity of the UV source used. 

4....The use of ZnO may results into leaching of Zn ions, author need to discuss how was the ZnO prepared to be stable and reusable.

5....The role of reactive radicals from the ZnO activation need to be discussed in detail using different scavenger studies to confirm its yield as well as reactivity with the dye. Author can take help from these studies, e.g., Chemical Engineering Journal 351 (2018) 841-855; Chemical Engineering Journal 356 (2019) 199-209;    

Author Response

Comment 1: The introduction is very poorly written, and the author needs to discuss the importance of this study, authors can take help from the following studies, e.g., Chemical Engineering Journal 351 (2018) 841-855: Scientific Reports 10 (2020), 2042; J. Clean. Prod. 241 (2019) 118263.

Reply 1: The introduction section was improved as recommended and refs were cited.

Comment 2: Why did the author select the green synthesis method for the synthesis of ZnO, though it is very tedious, time-consuming, and results in a low yield of ZnO?

Reply 2: Chitosan is a well-known biopolymer for the synthesis of metal and metal oxide nanoparticles. Chitosan was used as a capping agent during sol-gel synthesis of ZnO NPs and during annealing to control the size and morphology of the nanoparticles.

Comment 3: What was the intensity of the UV source used?

Reply 3: 1000 W

Comment 4: The use of ZnO may result in the leaching of Zn ions, author needs to discuss how was the ZnO prepared to be stable and reusable.

Reply 4: we have added more information to the article

Zinc ions (Zn2+) exhibit antimicrobial activity against various strains of bacteria and fungi. Partial dissolution of ZnO NPs releases Zn2+ ions in an aqueous suspension, which contributes to the antimicrobial activity of ZnO NPs. In addition to the soluble zinc species activity common to water-soluble zinc salts, ZnO combines two additional mechanisms of antimicrobial activity that complement its activity as a preservative in topical formulations: The generation of reactive oxygen species and direct contact with cell walls.

Comment 5:  The role of reactive radicals from the ZnO activation needs to be discussed in detail using different scavenger studies to confirm its yield as well as reactivity with the dye. The author can take help from these studies, e.g., Chemical Engineering Journal 351 (2018) 841-855; Chemical Engineering Journal 356 (2019) 199-209.

Reply 5: we added more info to the article and cited the refs

Photocatalysts are materials that break down hazardous organic compounds when exposed to sunlight. The valuable photocatalytic mechanism depends on the photocatalyst and configuration of the active species generated in the reaction medium. Radical scavenging experiments have been conducted in the literature to investigate the potential contribution of different active species to the degradation of AZO dyes. Radical scavengers such as isopropanol, formic, oxalic acid, and ascorbic acid were found to play an essential role in the degradation of AZO dyes under solar irradiation. Some of these radicals can potentially interact with the dyes due to their reduction potential therefore, a critical interpretation must be made when these species are added to a heterogeneous photocatalysis process. However, no radical scavengers were used in this study.

Reviewer 2 Report

Manuscript titled “Biosynthesis of Zinc Oxide Nanoparticles Using Different Chitosan Sources: Effects on Photocatalytic Degradation of AZO Dye and Antibacterial Activity”. In this study, an efficient and sustainable strategy for the biosynthesis of ZnO NPs from aqueous zinc chloride (ZnCl2) and a variety of chitosan sources (shrimp shell, crab shells and Streptomyces griseus) is proposed. This study has instructive significance in photocatalytic degradation of AZO Dye, but this manuscript needs major revisions to improve the quality of the manuscript.

1.       P2, line 66 to 78; The author should elaborate in the introduction why the three different chitosan are selected.

2.       P3, line 141 to 157. The author should simplify the description of catalyst crystallinity and crystallite size in subsection 3.1, which is too cumbersome.

3.       P5, line 172. Please confirm whether 45-75nm in Section 3.2 is 35-75nm.

4.       P5, line 173 to 178. The author should supplement EDS data of three catalysts in subsection 3.2(It is given in the form of figure or table).

5.       P6, line 183 to 185, Please confirm whether the absorption band data of three kinds of chitosan in Section 3.3 are correct.

6.       P6, line 190. The author should supplement the judgment standard of band gap value in subsection 3.3, and give its specific calculation method and formula.

7.       P7, line 208. In subsection 3.5, has the author considered the horizontal comparative analysis of the toxicity of ZnO nanoparticles made of three different kinds of chitosan to bacteria.

8.       P9, line 240. Did the author measure the content of free radicals in the photocatalytic process of the reaction system in the mechanism section of subsection 3.6? This can help to verify the author's analysis of the reaction mechanism.

9.       The author should carefully check the format of the whole paper and optimize the picture quality.

Author Response

Comment 1: P2, lines 66 to 78; The author should elaborate in the introduction on why the three different chitosan is selected.

Reply 1:  More information was added to the article “In this study, an efficient and green biosynthesis of ZnO NPs from aqueous zinc chloride (ZnCl2) and different type of chitosan (sources: shrimp shells, crab shells, and Streptomyces griseus) is proposed. The source of chitosan and the extraction process determines the molecular weight and degree of deacetylation of chitosan, which significantly affects the functionality and antibacterial activity of chitosan.”

Comment 2: P3, lines 141 to 157. The author should simplify the description of catalyst crystallinity and crystallite size in subsection 3.1, which is too cumbersome.

Reply 2: This section was rewritten. The XRD pattern of the predestined ZnO was used as a reference. Compared to the JCPDS cards, the diffraction peaks labeled (100), (002), (101), (102), (110), (103), (200), (112), (201), (004), and (202) were a good fit to the hexagonal ZnO wurtzite structure (JCPDS card no: 01-079-0205) [16]. The Debye-Scherer equation, D=Kλ/(βcosθ), was used to calculate the size of ZnO NPs' crystallites, where D is the average crystallite size, K is the form factor (0.9),  is of the wavelength of X-ray (0.15418 nm, CuKα),  is the maximum width at full width (FWHM) and is the Bragg angle [17]. Table 1 shows the XRD calculations (crystallite size and lattice parameters) of prepared ZnO NPs. The crystallite size of ZnO NPs varied between 30.9 to 35.8 nm.

Comment 3: P5, line 172. Please confirm whether 45-75nm in Section 3.2 is 35-75nm.

Reply 3: the value was corrected to 35-75 nm

Comment 4: P5, line 173 to 178. The author should supplement EDS data of three catalysts in subsection 3.2

Reply 4: The SEM-EDX was given in Figure and Table.

Compound

Composition of ZnO

ZnO NPs by shrimp shells

Element

Atomic percentage %

O  K

52.01

Zn  K

47.99

Totals

100

ZnO by crab shells

C K

12.43

O  K

52.59

Ne K

0.82

Zn  K

34.16

Totals

100

ZnO NPs by Streptomyces griseus bacteria

C K

22.07

O K

51.13

Zn K

26.8

Totals

100

Comment 5: P6, line 183 to 185, Please confirm whether the absorption band data of three kinds of chitosan in Section 3.3 are correct.

Reply 5: The sentence was modified. ZnO NPs synthesized using chitosan from shrimp shells, crab shells, and Streptomyces griseus bacteria show distinct absorption bands at 343, 330, and 328 nm, respectively.

Comment 6: P6, line 190. The author should supplement the judgment standard of band gap value in subsection 3.3, and give its specific calculation method and formula.

Reply 6: The paragraph was modified. UV-Vis spectra of the biosynthesized ZnO NPs are shown in Figure 3a. The UV-Vis spectra of the ZnO NPs synthesized with chitosan from crab shells, crab shells and Streptomyces griseus bacteria show distinct absorption bands at 343, 330 and 328 nm, respectively. The absorption spectrum of ZnO NPs is in agreement with that of Vijayakumar et al [20] and Vaseem et al. [21]. ZnO nanostructures are well-known photocatalysts that are generally active in UV light due to their wide band gap. It has been reported that the generated optical band gaps of ZnO nanostructures range from 2.7 eV to 4.7 eV with respect to the ZnO nanostructures formed in different solvent systems [12, 22, 23]. The efficiency of a ZnO photocatalyst strongly depends on its electronic band structure and band gap energy. The band gap energy for an efficient photocatalyst should be less than 3 eV to extend light absorption into the visible range and to efficiently utilize solar energy. The band gap energy of biosynthesized ZnO NPs was calculated by plotting (hv)2 versus energy (eV) as shown in Figure 3b. The band gap energies of ZnO NPs with chitosan from shrimp shells, crab shells and Streptomyces griseus bacteria are 2.73, 2.87 and 3.21 eV, respectively.

Comment 7: P7, line 208. In subsection 3.5, has the author considered the horizontal comparative analysis of the toxicity of ZnO nanoparticles made of three different kinds of chitosan to bacteria.

Reply 7: The comparative analysis of the toxicity of ZnO nanoparticles have not been considered for this study but it will be useful we consider this test in the future publications.

Comment 8: P9, line 240. Did the author measure the content of free radicals in the photocatalytic process of the reaction system in the mechanism section of subsection 3.6? This can help to verify the author's analysis of the reaction mechanism.

Reply 8: We didn't measure it but it will be useful if we consider this test in future publications.

Comment 9: The author should carefully check the format of the whole paper and optimize the picture quality.

Reply 9: The full article was revised.  

Reviewer 3 Report

Manuscript number: catalysts-2057871

Title: Biosynthesis of zinc oxide nanoparticles using different chitosan sources: Effects on photocatalytic degradation of AZO dye and antibacterial activity

The paper comprises only a description of figures without providing any physical explanations (discussions and interpretations). Furthermore, it is clear that the Photocatalytic part is problematic, and it should be greatly revised. It needs major rectifications/additions and clarifications before one can take a final decision:

1/ The paper contains several grammatical errors and typo mistakes that should be corrected. The English language should be greatly improved.

2/ The Abstract part should be greatly improved. It should clearly inform the important findings of the present study. It should also contain both qualitative and quantitative results.

3/ XRD part should be greatly improved: Kindly remove the 2theta angles and hkl values from the text and insert them only in XRD patterns. The definition of the parameters in the Scherrer equation should be rechecked, etc…

4/ The structural properties should be further discussed and greatly improved.

5/ The XRD patterns should be refined using the Rietveld method. The relevance factors (chi^2, RB, etc) of XRD patterns fitting should be added.

6/ What are the plausible reasons for the variations in lattice parameters? Also, what are the plausible reasons for the variations in crystallite size? 

7/ SEM image in Figure 2b showed an inhomogeneous shape distribution of particles. What is the plausible reason for that?

8/ The authors stated that “The preparation of ZnO NPs was confirmed by the study of SEM-EDX.”. Where are the EDX spectra? The authors should add the EDX spectra and atomic %.

9/ Add the Y-axis values in Figure 3. No details were provided for how to determine the band gap values!? No physical explanation is found for the difference in band gap energy values!?

10/ There are additional vibration bonds in FTIR spectra that should be identified.

11/ Add descriptions of the mechanisms of anti-bacterial activities as well as of photocatalytic activities.

12/ During an irradiation time of 60 min, it is clear that all samples do NOT show good photocatalytic activity. The values of degradation (%) should be majorly revised. It is impossible that these absorbance spectra reflect a >90% of degradation. They are greatly lower than 90%. You may need 2 or 3 hours to reach near full degradation with the current samples. Also, at 0 min, you have more than 80% of degradation!!? The photocatalytic part should be greatly revised.

I would like to see the revised manuscript. The paper should be sent to me for the second analysis after the major revisions. If the authors succeed to give satisfying answers to the above questions and suggestions and performing efficiently the required corrections, this work could be accepted for publication in this Journal.

Author Response

Comment 1: The paper contains several grammatical errors and typo mistakes that should be corrected. The English language should be greatly improved.

Reply 1: The English language was improved

Comment 2: The Abstract part should be greatly improved. It should clearly inform the important findings of the present study. It should also contain both qualitative and quantitative results.

Reply 2: The abstract was improved

Comment 3: XRD part should be greatly improved: Kindly remove the 2theta angles and hkl values from the text and insert them only in XRD patterns. The definition of the parameters in the Scherrer equation should be rechecked, etc…

Reply 3: This section was improved and carefully checked. Compared to the JCPDS cards, the diffraction peaks labeled (100), (002), (101), (102), (110), (103), (200), (112), and (201) are in a good fit to the hexagonal ZnO wurtzite structure (JCPDS card no: 01-079-0205) [22]. No other peaks are observed, indicating the high purity and crystallinity of synthesized ZnO NPs. The Debye-Scherer equation, D=Kλ/(βcosθ), was used to calculate the size of ZnO NPs' crystallites, where D is the average crystallite size, K is the form factor (0.9),  is of the wavelength of X-ray (0.15418 nm, CuKα),  is the maximum width at full width (FWHM) and is the Bragg angle [23].

Comment 4: The structural properties should be further discussed and greatly improved.

Reply 4: This section was improved. SEM, UV-vis, and XRD sections were improved

Comment 5: The XRD patterns should be refined using the Rietveld method. The relevance factors (chi^2, RB, etc) of XRD patterns fitting should be added.

Reply 5:  This section was improved, the fitting is done by XRD software combined with the instrument and the crystallite sizes are given as obtained from the software.

Comment 6: What are the plausible reasons for the variations in lattice parameters? Also, what are the plausible reasons for the variations in crystallite size?

Reply 6:  Table 1 shows the XRD calculations (crystallite size and lattice parameters) of prepared ZnO NPs. The crystallite size of ZnO NPs varied between 30.9 to 35.8 nm. Different prop-erties of chitosan (capping agent) lead to a different degree of stabilization of Zn(OH)2 NPs against agglomeration. This leads to the formation of ZnO NPs with the observed variation in crystallite size and lattice parameters.

Comment 7: SEM image in Figure 2b showed an inhomogeneous shape distribution of particles. What is the plausible reason for that?

Reply 7: Different sources of chitosan lead to a different degree of stabilization of Zn(OH)2 NPs against agglomeration. This leads to the formation of ZnO NPs with the observed variation in particle sizes. The crystallite size is smaller than the particle size determined by the SEM analysis. This may indicate that the prepared ZnO NPs are polycrystalline.

Comment 8: The authors stated that “The preparation of ZnO NPs was confirmed by the study of SEM-EDX.”. Where are the EDX spectra? The authors should add the EDX spectra and atomic %.

Reply 8: Spectra and table was added to the article

Comment 9: Add the Y-axis values in Figure 3. No details were provided for how to determine the band gap values!? No physical explanation is found for the difference in band gap energy values!?

Reply 9: More information was added.

ZnO NPs are well-known photocatalysts that are generally active in UV light due to their wide band gap. The efficiency of ZnO NPs strongly depends on their electronic band structure, UV-Vis absorption, and band gap energy. It has been reported that the generated optical band gaps of ZnO nanostructures range from 2.7 eV to 4.7 eV for ZnO NPs formed in different solvent systems [26, 27]. The bandgap energy for an efficient photocatalyst should be less than 3 eV to extend light absorption into the visible range and efficiently utilize solar energy. The UV-Vis spectra of the ZnO NPs synthesized from different chi-tosan sources are shown in Figure 4a. The UV-Vis spectra of the ZnO NPs synthesized from different chitosan sources (crab shells, crab shells and Streptomyces griseus bacteria) show distinct absorption bands at 343, 330 and 328 nm, respectively. The absorption spectrum of the ZnO NPs is in agreement with that of Vijayakumar et al[28] and Vaseem et al. [29]. The band gap energy of synthesized ZnO NPs was calculated by plotting (hv)2 versus energy (eV) as shown in Figure 4b. The band gap energies of ZnO NPs with chi-tosan from shrimp shells, crab shells and Streptomyces griseus bacteria are 2.73, 2.87 and 3.21 eV, respectively. These results confirm that the use of chitosan during the ZnO NPs synthesis has significantly affected the optical properties of the ZnO NPs and reduced the bandgap energy to a minimum. 

Comment 10: There are additional vibration bonds in FTIR spectra that should be identified.

Reply: FTIR made before annalesing at 500 C so we belive it do not add much information about the final product so we deleted the curve.

Comment 11: Add descriptions of the mechanisms of anti-bacterial activities as well as of photocatalytic activities.

Reply 11: we have discussed both the anti-bacterial activities as well as of photocatalytic activities.

Comment 12: During an irradiation time of 60 min, it is clear that all samples do NOT show good photocatalytic activity. The values of degradation (%) should be majorly revised. It is impossible that these absorbance spectra reflect a >90% of degradation. They are greatly lower than 90%. You may need 2 or 3 hours to reach near full degradation with the current samples. Also, at 0 min, you have more than 80% of degradation!!? The photocatalytic part should be greatly revised.

 Reply 12:  we have repeated the experiment and corrected the calculation of degradation effiecny as recommended.

Round 2

Reviewer 1 Report

The manuscript is recommended for acceptance subject to citing both the recommended references. 

Reviewer 3 Report

The revised manuscript has been greatly improved. Furthermore, the results of degradation % seem now correct. I think that the revised manuscript could be now accepted for publication.